# The Upstream Pathway of mTOR-Mediated Autophagy in Liver Diseases

**DOI:** 10.3390/cells8121597

**Published:** 2019-12-09

**Authors:** Haojie Wang, Yumei Liu, Dongmei Wang, Yaolu Xu, Ruiqi Dong, Yuxiang Yang, Qiongxia Lv, Xiaoguang Chen, Ziqiang Zhang

**Affiliations:** 1College of Animal Science and Technology, Henan University of Science and Technology, Luoyang 471000, China; haojiewang@stu.haust.edu.cn (H.W.); 180218250878@stu.haust.edu.cn (Y.X.); 180218250881@stu.haust.edu.cn (R.D.); yyxiang2018@126.com (Y.Y.); lvqx20001@163.com (Q.L.); cxguang1015@126.com (X.C.); 2College of Medical, Henan University of Science and Technology, Luoyang 471000, China; zzq828929@163.com

**Keywords:** autophagy, AKT, AMPK, ERK, liver diseases, mTOR, MEK, PI3K, Ras, Raf

## Abstract

Autophagy, originally found in liver experiments, is a cellular process that degrades damaged organelle or protein aggregation. This process frees cells from various stress states is a cell survival mechanism under stress stimulation. It is now known that dysregulation of autophagy can cause many liver diseases. Therefore, how to properly regulate autophagy is the key to the treatment of liver injury. mechanistic target of rapamycin (mTOR)is the core hub regulating autophagy, which is subject to different upstream signaling pathways to regulate autophagy. This review summarizes three upstream pathways of mTOR: the phosphoinositide 3-kinase (PI3K)/protein kinase (AKT) signaling pathway, the adenosine monophosphate-activated protein kinase (AMPK) signaling pathway, and the rat sarcoma (Ras)/rapidly accelerated fibrosarcoma (Raf)/mitogen-extracellular activated protein kinase kinase (MEK)/ extracellular-signal-regulated kinase (ERK) signaling pathway, specifically explored their role in liver fibrosis, hepatitis B, non-alcoholic fatty liver, liver cancer, hepatic ischemia reperfusion and other liver diseases through the regulation of mTOR-mediated autophagy. Moreover, we also analyzed the crosstalk between these three pathways, aiming to find new targets for the treatment of human liver disease based on autophagy.

## 1. Introduction

The liver is the largest solid organ of the human body, which plays a key pivotal role in many physiological processes such as nutrient storage and metabolism, synthesis of new molecules and purification of toxic chemicals [1]. A variety of factors, such as chemical contaminants, viruses, drugs, and alcohol, can disrupt the above-mentioned normal functions of the liver and cause hepatic steatosis, hepatitis, fibrosis, cirrhosis, liver cancer, and other liver diseases, which are seriously harmful to human health. Autophagy is a process of lysosomal degradation that regulates the homeostasis of organelles and proteins. As a cellular housekeeper, the function of autophagy is mainly divided into two types: the turnover of old molecules or damaged molecules and the supplement of nutrient storage during starvation.

Accumulated studies have demonstrated that autophagy plays a crucial role in regulating liver physiology and balancing liver metabolism [2]. Additionally, autophagy is also involved in the occurrence and development of liver diseases mentioned above. In one aspect, autophagy protects liver cells from damage and cell death by eliminating damaged organelles and proteins introduced in liver-related diseases [2]. On the other hand, under different conditions, autophagy can promote further deterioration of liver injury (e.g. excessive autophagy can cause autophagic cell death of hepatocytes; increasing autophagy of hepatic stellate cells can promote its activation and aggravate hepatic fibrosis) [2,3]. Therefore, how to properly regulate autophagy in different situations becomes very important in the treatment of liver injury.

Of note, it is well documented that mechanistic target of rapamycin (mTOR) plays a pivotal role in autophagy regulation. mTOR plays a negative role in autophagy by regulating autophagy related proteins and lysosome biosynthesis. Importantly, mTOR is subject to a variety of different upstream signaling pathways, which can correspondingly inhibit or enhance autophagy levels by regulating mTOR. Thus, the regulation of different upstream signaling pathways of mTOR may be a new research idea for the treatment of liver injury. In this review, we analyzed the role of several different upstream pathways mediated autophagy of mTOR in different liver injuries, and summarized the crosstalk between several upstream pathways of mTOR. It will provide some new therapeutic targets for treating liver injury by regulating autophagy.

## 2. Autophagy

Autophagy is a process by which cells degrade and metabolize their own components, which is divided into non-selective autophagy and selective autophagy. Non-selective autophagy is used for the turnover of bulk cytoplasm under starvation conditions while selective autophagy specifically targets damaged or excess organelles, including damaged mitochondria, unneeded peroxisomes, excess ribosomes and lipid droplets, as well as invasive microorganisms [4]. Under basic conditions, all cells have lower levels of autophagy, and can be further induced by different forms of stress such as nutrient or energy starvation, growth factor depletion, infection and hypoxia [5]. Based on the method of target substance capture and delivery to lysosome, this evolutionarily highly conserved process can be separated into macroautophagy, microautophagy and chaperone-mediated autophagy (CMA). CMA uses chaperones to identify cargo proteins containing specific pentapeptide motifs and then translocated directly across the lysosomal membrane [6]. By contrast, macroautophagy and microautophagy involve dynamic membrane rearrangement in capturing cargo. Microautophagy directly ingests cargo through lysosomal limiting membrane invagination, protrusion and separation [7]. Macroautophagy uses a special cytoplasmic vesicle to capture cargo and eventually degrade with lysosomes [5] (Figure 1). At present, most of our knowledge focuses on macroautophagy (hereinafter referred to as autophagy), selective macroautophagy in mammalian cells is the focus of this review.

### 2.1. Membrane Rearrangement of Autophagy Process

Upon induction, a membrane structure called phagophore will extend and isolate cytoplasmic organelles, including the endoplasmic reticulum, ribosomes, and mitochondria [3] (Figure 1). As the phagophore expands, the membrane bends and eventually the edges of the membrane fuse to form a bilayer membrane structure, autophagosome, which is the key event in autophagy [3]. However, the source of membranes involved in the formation of phagophore and autophagosome remains unclear. It is possibly the endoplasmic reticulum, mitochondria, Golgi apparatus and plasma membrane, although it is now widely believed that the endoplasmic reticulum is the main source of the membrane [8,9,10]. Once the autophagosome is formed, it must transport its cargo to the lysosome [5]. When it reaches its destination, the outer membrane of the autophagosome will fuse with the lysosomal membrane [5]. Subsequently, the lysosomal hydrolase degrades the inner membrane of the autophagosome to form a monolayer membrane structure, autolysosome. At the same time, the cytoplasmic organelles contained in the autophagosome are also degraded by hydrolases. The amino acids and other small molecules produced by the degradation are transported back to the cytoplasm for recycling or for energy production [11].

### 2.2. Molecular Machinery of Autophagy

The above autophagy process can usually be subdivided into different stages: initiation, nucleation of the autophagosome, expansion and elongation of the autophagosome membrane, closure and fusion with the lysosome, and the degradation of intravesicular products [12] (Figure 1). The different steps of autophagy involve different evolutionary conserved autophagy-related genes (ATGs) and the mechanisms behind them are complex.

Briefly, the initiation phase begins with the activation of the uncoordinated 51-like kinase (ULK) complex (comprising ULK1, ULK2, ATG13, Focal adhesion kinase family interacting protein of 200 Kd (FIP200) and ATG101) [12,13,14,15,16]. When ULK1/2 is phosphorylated, it interacts with ATG13, which directly binds to FIP200 and mediates the interaction of FIP200 with ULK1/2 [13,17]. Furthermore, ATG101 can stabilize the function of ULK complex by directly binding to ATG13 [15,18]. Next, the activated ULK complex recruits a class III phosphatidylinositol 3 Kinase (PI3K) complex I that includes Beclin-1, vacuolar protein sorting 34 (VPS34) that converts PI into PI-3-phosphate (PI3P), general vesicular transport factor (p115), ATG14 and activating molecule in Beclin-1-regulated autophagy protein 1 (AMBRA1) to the autophagosome forming site during the nucleation phase [19,20]. The important role of Beclin-1 in regulating class III PI3K complex I is essential for autophagy, which can activate or inhibit autophagy by binding to some proteins [21]. For example, the anti-apoptotic protein B-cell lymphoma 2 (Bcl-2) or B-cell lymphoma extra-large (Bcl-X_L_) can bind to the BH3 domain in Beclin-1, inhibiting its interaction with VPS34, thereby inhibiting autophagy [22,23,24]. On the other hand, activating molecule in Beclin-1-regulated autophagy protein 1 (AMBRA1) can bind to Beclin-1, enhancing the interaction between Beclin-1 and VPS34, thus activating autophagy [25]. When the class III PI3K complex I is activated, which in turn activates local phosphatidylinositol-3-phosphate (PI3P) production at a characteristic ER structure called the omegasome [20]. From its inception at the omegasome, the phagophore elongates into a cup-shaped structure and begins to engulf cellular material. Subsequently, PI3P recruits the PI3P effector proteins tryptophan-aspartic acid (WD) repeat domain phosphoinositide-interacting proteins (WIPI2) and zinc-finger FYVE domain-containing protein 1 (DFCP1) to the omegasome. Recently WIPI2 has been shown to bind directly to ATG16L1 and promote the formation of ubiquitin-like protein binding systems [26]. These events are accompanied by the recruitment of ATG9-containing vesicles generated, which may deliver additional lipids and proteins contributing to membrane expansion [27,28].

In the elongation/enclosure phase of the phagophore membrane, two ubiquitin-like protein binding systems are involved: the ATG12-ATG5-ATG16L1 complex and the microtubule-associated protein 1 light chain 3 (LC3) –phosphatidylethanolamine (PE) conjugate [29]. In the first ubiquitin-like reaction, the ubiquitin-like protein ATG12 was first activated by ATG7 (E1 ubiquitin-activating enzyme-like), then transferred to ATG10 (E2 ubiquitin-binding enzyme-like), and finally covalently connected to the internal lysine of the ATG5 to form the ATG12-ATG5 conjugate [30,31]. Subsequently, the ATG12-ATG5 conjugate interacts with a coiled-coil protein ATG16L1, which binds the ATG12-ATG5-ATG16L1 complex into a tetramer by self-oligomerization and is eventually recruited into the phagophore by the WIPI2 protein. [32,33,34]. The second ubiquitin-like reaction involves the LC3 (ATG 8 homolog) which can mediate membrane tethering and hemifusion plays a key role in the process of phagophore membrane expansion and eventual closure as autophagosomes [30,35]. First, LC3 is cleaved at the C-terminus by a cysteine protease ATG4, exposing a glycine residue, forming the cytoplasmic isomer LC3-I [5,30,36,37]. Subsequently, LC3-I is activated by the E1-like enzyme ATG7 and transferred to the E2-like enzyme ATG3 [38,39]. Finally, the C-terminal glycine of LC3-I is covalently bound to PE (ATG12-ATG5-ATG16L1 complex may be used as E3-like ligase to facilitate this last step) [38,40]. In the manner described above, LC3 was transformed from free diffusion form (LC3-I) to membrane anchored lipidated form (LC3-II) [20,41]. Remarkably, LC3-II is specifically designed for elongated autophagosome membranes and is located on the inner and outer membranes [30]. When autophagosomes are fused to lysosomes, LC3-II located on the autophagosomal inner membrane is degraded within the autolysosome, whereas LC3-II located on the cytoplasmic face of autolysosome is destroyed by ATG4 and recycled [42]. LC3-II is widely considered to be a good marker for studying autophagy due to this specific binding of LC3-II to autophagosome membranes [30,43].

In the final stages of the autophagy process: fusion and degradation. A large number of factors regulate this key process [44]. For example, in mammalian cells, fusion regulators include Ras-related proteins in brain small GTPases (such as Rab7 and Rab11), soluble N-ethylmaleimide-sensitive factor attachment protein receptors (SNARES, such as syntaxin-17, SNAP-29, and VAMP8), homotypic fusion and vacuole protein sorting (HOPS) complex components (such as Vps16, Vps33A, and Vps39), endosomal sorting complexes required for transport (ESCRT) components (such as SKD1, CHMP2B, and CHMP4B), FYVE and coiled-coil domain-containing protein (FYCO1), lysosome-associated membrane protein type 2 (LAMP-2), and lysosomal acid hydrolases include cathepsin B, D, L [44,45,46,47,48]. However, with regard to the respective roles of these factors, we need further research.

## 3. Overview of mTOR Signaling Pathway

### 3.1. mTOR Structure

mTOR is a highly conserved serine/threonine protein kinase in the PI3K-related kinase (PIKK) family that consists of 2549 amino acids and comprises multiple functional as well as regulatory domains [49,50] (Figure 2A). The N-terminal contains 20 tandem Huntingtin-Elongation factor 3-regulatory subunit A of PP2A-TOR1 (HEAT) repeats involved in protein-protein interactions [51]. The C-terminal contains the protein kinase (KIN) domain, and its upstream is followed by the FKBPl2-Rapamycin Binding (FRB) domain and the focal adhesion targeting (FAT) domain, which forms a spatial structure with the FATC (another FAT domain located at the C-terminal end) and exposes the mTOR molecular catalytic domain by a FAT/FATC method combined with each other [49,52,53,54]. In addition, mTOR also contains a negative regulatory domain (NRD) between the catalytic domain and the FATC domain, which is essential for the regulation of mTOR activity [54].

### 3.2. Molecular Composition of mTORC1 and mTORC2 

mTOR can recruit multiple chaperones through the above different relatively independent domains to form two distinct signaling complexes: mTOR complex 1 (mTORC1) and mTOR complex 2 (mTORC2) (Figure 2B). mTORC1 consists of three core components: mTOR, regulatory protein associated with mTOR (Raptor), mammalian lethal with Sec13 protein 8 (mLST8), and two inhibitory subunits: proline-rich Akt substrate of 40 kDa (PRAS40), DEP domain containing mTOR interacting protein (Deptor) [55,56,57,58,59,60]. Among them, mTOR is the catalytic subunit of the entire complex. Raptor is a mTOR regulatory related protein that binds to the mTOR signaling motif to promote substrate recruitment of mTORC1 [61,62]. mLST8 is associated with the mTORC1 catalytic domain and stabilizes kinase activation [63]. PRAS40 and Deptor are the two negative regulatory subunits of mTORC1.When the activity of mTORC1 is decreased, PRAS40 and Deptor will be recruited by mTOR, further inhibiting mTORC1 expression; when mTORC1 is activated, phosphorylation of PRAS40 and Deptor directly reduces their inhibition and further activates mTORC1 signaling [59,64,65,66]. Rapamycin, a macrolide antibiotic originally purified from Streptomyces hygroscopicus [67], can inhibit mTORC1 activity by interacting with the intracellular receptor FK506-Binding Protein (FKBP12) to form a drugprotein complex that binds to the FRB domain of mTOR [53,63,68]. However, mTORC2 is characterized by insensitivity to acute rapamycin therapy [60]. Like mTORC1, mTORC2 also contains the mTOR catalytic subunit, mLST8 and Deptor [60,69]. In addition to these protein components, mTORC2 also contains rapamycin insensitive companion of mTOR (Rictor) [70,71], mammalian stress-activated protein kinase interacting protein (mSIN1) [72,73] and protein observed with rictor1 and 2 (Protor1/2) [74,75,76]. Although the rapamycin-FKBP12 complex does not directly inhibit mTORC2, prolonged rapamycin treatment also reduces the integrity and activity of mTORC2, probably because the interaction between Rictor and mTOR is prevented during the formation of new mTORC2 [60,77].

### 3.3. Cellular Processes Downstream of mTORC1 and mTORC2

These two kinase complexes have specific substrate preferences, so they can trigger different downstream signaling events to regulate cell function [78] (Figure 2B). The main function of mTORC1 is to regulate cell growth and metabolism. It integrates a variety of stimulation and signaling networks to promote anabolism. For example, mTORC1 can promote protein synthesis by directly phosphorylating two key substrates, phosphorylating 70 kDa ribosomal protein S6 kinase 1 (p70S6K1) and the eukaryotic translation initiation factor 4E binding protein (4EBP) [60]. mTORC1 can promote novel lipid synthesis through the sterol response element binding protein (SREBP) transcription factor to satisfy membrane formation during cell growth [79]. Furthermore, mTORC1 also promotes the synthesis of nucleotides required for DNA replication and ribosome synthesis in cell growth and proliferation [60,80,81]. In addition to promoting the above anabolic processes, mTORC1 also negatively regulates the catabolic process-autophagy, which promotes cell growth by inhibiting protein catabolism. The specific regulatory mechanism will be discussed in the next section. Compared to mTORC1, mTORC2 controls cell proliferation, survival, ion transport and cytoskeletal remodeling primarily through phosphorylation of several members of the protein kinase A, G and C (AGC) family [82]. For example, mTORC2 can phosphorylate protein kinase C (PKC)α [70,71], PKCδ [83], PKCζ [84], PKCγ and PKCε [85] to regulate cytoskeletal remodeling and cell migration. Furthermore, mTORC2 also phosphorylates AKT. Activated AKT promotes cell survival, proliferation and growth by acting on several key substrates, including Forkhead box O1/3 (FoxO1/3a) transcription factor, glycogen synthase kinase 3β (GSK3β) and tuberous sclerosis 2 (TSC2) protein [72,82,86]. Moreover, mTORC2 also phosphorylates and activates serum/glucocorticoid-regulated kinase 1 (SGK1), which regulates ion transport and cell survival [87].

### 3.4. Upstream Regulation of mTORC1 and mTORC2

Importantly, the above effects of mTORC1/2 are subject to a variety of upstream signals (Figure 2B). The mTORC1 pathway integrates at least four major intracellular and extracellular signals: growth factors, energy status, oxygen content and amino acids [60]. The most important protein involved in the regulation of mTORC1 activity is the Tuberous Sclerosis Complex (TSC), a heterotrimeric complex composed of TSC1, TSC2 and Tre2-Bub2-Cdc16 (TBC) 1 domain family, member 7 (TBC1D7) [88], and functions as a GTPase activating protein (GAP) for the small GTPase Rheb [60]. Since only the GTP-bound form of Rheb can directly bind and activate mTORC1. Therefore, as a Rheb GAP, TSC suppresses mTORC1 by converting Rheb from an active GTP-bound form to an inactive GDP-bound state [69,89,90]. Growth factor regulates mTORC1 activity mainly through activation of two classical upstream pathways of mTORC1: PI3K/AKT/mTORC1 and Ras/Raf/MEK/ERK/mTORC1 signaling pathways [60] (specific regulatory mechanisms are described in more detail below). Oxygen level affects mTORC1 activity in many ways. Hypoxia can activate TSC through transcriptional regulation of regulated in DNA damage and development 1 (REDD1) to block mTORC1 signal transduction [91]. In addition, under mild hypoxic conditions, the decrease of ATP level activated AMPK, also promoted TSC activation and inhibited mTORC1 signal transduction. Similarly, the energy state of the cells also activates the AMPK/mTORC1 signaling pathway by altering ATP levels, thereby modulating mTORC1 activity (see below). Furthermore, amino acids, particularly leucine and arginine, also activate mTORC1. Recently, there are two different ways in which amino acids regulate mTORC1: one is that leucine and arginine depend on Rag GTPase to activate mTORC1, the other is glutamine to activate mTORC1 independently of the Rag GTPases [82]. Compared to mTORC1, much less is known about the mTORC2 pathway. The only known upstream activating factor is the growth factor/PI3K signal axis [78]. One study suggested that PI3K signaling may activate mTORC2 by promoting the binding of mTORC2 to ribosomes [92]. However, the mechanism of how ribosome binding activates mTORC2 needs further investigation.

### 3.5. The Role of mTOR in Liver Metabolism and Liver Disease

Consistent with the role of mTOR signaling in coordinating anabolism and catabolism at the cellular level, studies have shown that mTOR signaling is also necessary for metabolic regulation in the organisms [60]. The liver is an important organ of systemic metabolism that plays a key role in regulating ketones, lipid metabolism, systemic glucose and insulin homeostasis [93,94,95]. Evidence suggests that mTOR plays an important regulatory role in these processes. Upon fasting, the liver performs multiple functions to maintain a systemic balance, including increasing the production of ketone bodies, which provide energy to peripheral tissues [96]. Sengupta et al. [96] found that mTORC1 is activated in the liver of mice with liver-specific loss of TSC1 (L-TSC1 KO), resulting in significant defects in ketone body production and ketogenic gene expression during fasting, whereas the deletion of Raptor (a core component of mTORC1) can reverse these effects. It suggested that mTORC1 can control ketogenesis in mice in response to fasting. Lipogenesis is activated by transcription factor SREBP [97,98]. In the liver, mTORC1 can activate the expression and maturation of SREBP through two independent pathways. One is the S6K1-dependent manner that mediates the maturation of SREBP-1 [79,99]. Another one is independent of S6K1. Under the action of nutrients and growth factors, mTORC1 directly phosphorylates mTORC1 substrate phosphatidic acid phosphatase lipin-1 (a negative regulator of SREBP-1 activity) to prevent lipid-1 translocation from entering the nucleus, which makes SREBP transcriptionally active, ultimately stimulating neonatal lipid synthesis [100,101]. In addition, mTORC1 in the liver also plays a key regulatory role in systemic glucose and insulin homeostasis. In the L-TSC1 KO mice, chronic activation of the mTORC1 signal produces a strong inhibitory feedback on insulin receptor substrate 1 (IRS1), which causes a decrease in the AKT signaling pathway, leading to glucose intolerance [96,102,103]. In contrast, liver-specific Raptor-deficient mice showed higher systemic glucose tolerance, probably due to enhanced AKT signaling pathway and insulin-induced hepatic glucose uptake [100,104]. 

Similarly, the liver also needs mTORC2 to maintain the balance of lipid and glucose. For example, liver-specific knockout Ricto (a core component of mTORC2) mice exhibit hypolipidemic and hyperglycemia, and mechanistic studies suggest that AKT plays an important role in the regulation of homeostasis by mTORC2 [101,105,106,107]. In summary, all the above evidences demonstrate the important role of mTOR in the regulation of liver metabolism. In addition, mTOR also plays an important role in liver injury, and cumulative evidence has shown that mTOR-mediated autophagy to treat liver injury is a potential mechanism. For example, in hepatic ischemia-reperfusion injury, mTOR-mediated activation of hepatocyte autophagy can promote hepatocyte survival, inhibit hepatocyte death/apoptosis, and thereby repair hepatic ischemia-reperfusion injury [108]. In liver cancer, inhibition of mTOR-mediated hepatoma cell protective autophagy can accelerate the death of liver cancer cells, and thus alleviate the malignant development of liver cancer [109]. Therefore, mTOR-mediated autophagy may be a therapeutic target for liver injury. 

## 4. Regulation of Autophagy by mTOR

mTORC1 is a key regulator of cell catabolism, especially in autophagy. Accumulating evidences have shown that mTORC1 can negatively regulate autophagy. However, its mechanism of regulating autophagy is complex, and the specific ways of action have only been gradually clarified in recent years. In general, the regulation of autophagy-related proteins and lysosomal biogenesis is the key to the whole process. In mammalian cells, mTORC1 regulates autophagy in at least four ways (Figure 3). Firstly, under nutrient-rich conditions, mTORC1 inhibits the initiation of autophagy by inhibiting ULK complexes. In one aspect, mTORC1 phosphorylates ULK1 at Ser758, prevents the interaction between ULK1 and AMPK, and thus inhibits ULK1 activity [110]. AMPK is necessary for regulation of autophagy, during glucose starvation AMPK phosphorylates ULK1 at Ser317/Ser777 and activates it [110]. On the other hand, mTORC1 also directly phosphorylates ATG13 to inhibit the activity of the ULK complex, whereas the mTORC1 phosphorylation site in ATG13 remains to be determined [13,17,111]. Secondly, mTORC1 inhibits autophagy by directly phosphorylating AMBRA1 [112]. AMBRA1 promotes nucleation of autophagosomes by promoting the interaction of Beclin-1 with lipid kinase VPS34 [113]. In addition, AMBRA1 also plays a key role in stabilizing ULK1 and activating ULK1 kinase activity [112]. Interestingly, under nutrient-rich conditions, mTORC1 directly phosphorylates AMBRA1 and inhibits its function of activating ULK1, thereby inhibiting autophagy [68]. Thirdly, mTORC1 inhibits autophagy by directly phosphorylating ATG14 in the class III PI3K complex I at multiple sites (Ser3, Ser223, Thr233, Ser383, and Ser440) [114]. Phosphorylation of ATG14 inhibits the production of phosphatidylinositol-3-phosphate (PtdIns3P) by the related kinase VPS34, which is critical for the formation of autophagosomes [115]. Fourthly, mTORC1 can also indirectly inhibit autophagy by regulating the transcription of genes required for lysosomal biogenesis [116]. The transcription factor EB (TFEB) is a major transcriptional regulator of lysosomal biogenesis and autophagy genes [117]. Upon entry into the nucleus, TFEB promotes transcription of genes encoding proteins required for lysosomal biogenesis and autophagy, thereby indirectly activating autophagy [114]. mTORC1 directly phosphorylates the Ser142 and Ser211 sites of TFEB to retain it in the cytoplasm to inhibit autophagy [116,118,119]. In addition to the above four ways, mTORC1 can also inhibit the late phase of autophagy. For example, Kim et al. [120] reported that mTORC1 prevents the maturation of autophagosomes by phosphorylating the Ser498 site of ultraviolent irradiation resistance- associated gene (UVRAG). Cheng et al. [121] demonstrate that under nutrient-rich conditions, mTORC1 phosphorylates Pacer at serine157 to disrupt the association of Pacer with Stx17 and the HOPS complex and thus abolishes Pacer-mediated autophagosome maturation. Recently, Wan et al. [122] discovered a new target of mTORC1, acetyltransferase p300. Activated mTORC1 interacts with p300 and phosphorylates p300 at four serine residues in the C-terminal region to inhibit autophagy. WIPI2 as a critical protein in isolation membrane growth and elongation has also recently been identified as a key downstream substrate for mTOR regulation of autophagy. Wan et al. [123] reported that mTORC1 can phosphorylates Ser395 of WIPI2, directing WIPI2 to interact specifically with the E3 ubiquitin ligase HUWE1 for ubiquitination and proteasomal degradation. Thereby inhibiting the formation of autophagosome blocks the autophagy flux. The above illustrates the detailed mechanism by which mTORC1 regulates autophagy, however, it is not fully understood how mTORC2 regulates autophagy. Studies have shown that mTORC2 can phosphorylate AKT at the hydrophobic motif site Ser473, activating the AKT/mTORC1 signal axis [78]. Therefore, perhaps mTORC2 may indirectly inhibit autophagy by activating mTORC1. Whether mTORC2 can directly regulate autophagy remains to be further studied.

## 5. Effect of Autophagy on Liver Injury by Targeting Upstream Pathway of mTOR

As previously mentioned, mTOR is a major modulator of autophagy and it receives inputs from different signaling pathways, especially from those that sense the energetic state of the cell to trigger or halt the synthesis of proteins [124]. Different upstream pathways of mTOR play an important role in the regulation of autophagy, and a lot of them are associated with liver damage. So, the upstream pathway of mTOR is an appealing target of autophagy regulation in the treatment of liver injury. In this section we discuss several important upstream pathways of mTOR regulating autophagy and their significance in the treatment of liver injury (Figure 3).

### 5.1. PI3K/AKT/mTOR/Autophagy Signaling Pathway

The PI3K/AKT signaling pathway is involved in the regulation of multiple cellular functions such as proliferation, apoptosis and autophagy. PI3Ks are members of the intracellular lipid kinase family that phosphorylate the hydroxyl groups of 3′-phosphatidylinositol and phosphoinositides. They have different structures, functions and substrate preferences, so PI3Ks can be divided into three categories, class I, class II and class III [125,126]. Among these classes, class I PI3Ks play vital roles in regulating cell growth and survival [127]. So here we exclusively discuss signaling downstream of class I PI3Ks. The class I PI3Ks is a heterodimeric enzyme consisting of a regulatory subunit and a catalytic subunit. Class I PI3Ks can be activated by a variety of upstream pathways that link a wide range of cell surface receptors to specific PI3Ks isoforms, according to the characteristics of its connection, class I PI3Ks is classified into class IA PI3Ks and class IB PI3Ks [128]. Class IA PI3Ks are activated by receptor tyrosine kinases (RTKs), its catalytic subunit is p110 including three subtypes of p110α, p110β and p110δ, and the regulatory subunit is P85 including five subtypes of p85α, p55α, p50α, p85β and p55γ [128]. When the cell is stimulated by growth factors, the upstream receptor of the class IA PI3Ks will be activated and initiating a series of signaling cascades. The activated intracellular portion of the receptor may interact directly with the Src-homology 2 (SH2) domain of the regulatory subunit p85 or indirectly with P85 via adaptor molecules (such as IRS1) [129]. Then this interaction allows the activation of the catalytic subunit of class IA PI3Ks. While class IB PI3Ks are activated by G-protein-coupled receptors (GPCRs) and consists of a heterodimer of the catalytic subunit p110γ and the regulatory subunit p101 or its homolog p84 and p87PIKAP (PI3Kγ adaptor protein of 87 kDa) [128,130,131,132]. Compared to class IA PI3Ks, when the cells are stimulated, the regulatory subunit of class IB PI3Ks directly interacts with the Gβγ subunit of the trimeric G protein, and then the catalytic subunit of class IB PI3Ks are activated [128,133,134]. All these changes in class I PI3Ks eventually phosphorylates the inositol ring of the membrane phospholipid, phosphatidylinositol-4,5-bisphosphate (PI-4,5-P_2_), to generate phosphatidylinositol-3,4,5-trisphosphate (PI-3,4,5-P_3_) at the cytoplasmic face of the plasma membrane [135]. However this activity of class I PI3Ks can be offset by phosphatase and tensin homologue deleted on chromosome 10 (PTEN), which converts PI-3,4,5-P_3_ back to PI-4,5-P_2_, and inositol-5-phosphatases including SH2-domain containing inositol phosphatase (SHIP), which convert PI-3,4,5-P_3_ to phosphatidylinositol-3,4-bisphosphate (PI-3,4-P_2_) [136]. As the major negative regulator of the PI3K/AKT pathway, studies have shown that conditions of PTEN-deficient mice manifest as steatosis, hepatomegaly, fibrosis, and hepatocellular carcinoma [137,138]. Then the generated PI-3,4,5-P_3_ acts as a second messenger, recruits AKT (also known as protein kinase B) and phosphoinositide dependent protein kinase 1 (PDK1) to the cell membrane through their PH domains, where the amino acid residues of Ser473 and Thr308 of AKT are phosphorylated by the mTORC2 complex and protein PDK1, respectively, to complete its activation [139,140].

AKT belongs to the family of serine/threonine protein kinase conserved in mammals and consists of three distinct subunits, AKT1, AKT2 and AKT3. In the liver, AKT1 and AKT2 are the only expression forms in healthy state, of which AKT2 is most expressed in hepatocytes, accounting for 85% of total liver AKT [141]. AKT serves as a key downstream target for PI3K, and a central mediator of the PI3K pathway, which exerts multiple downstream effects by phosphorylating many substrates involved in cell survival, proliferation, metabolism, and movement [142]. Cumulative evidence suggests that the PI3K/AKT signaling pathway plays a role in liver injury. For instance, Kim et al. [143] found that carbon monoxide pretreatment can inhibit the activity of GSK3β, a downstream substrate of AKT, by inducing activation of the PI3K/AKT pathway in the liver, thereby reducing the expression of inflammatory factors, enhancing anti-inflammatory ability, and reducing the degree of hepatocyte damage. Recent studies have shown that hypothermia preconditioning can inhibit apoptosis and inflammatory damage by activating the PI3K/AKT/FoxO3a pathway, providing an effective means of protecting the liver [144]. In addition to GSK3 and the FoxO transcription factors, mTOR is also a key downstream signal node of AKT. The main mechanism by which AKT activates mTORC1 is through direct phosphorylation and inhibition of TSC2 (a potent inhibitor of mTORC1) [145]. The AKT-mediated phosphorylation of TSC2 abolished its inhibitory effect on mTORC1 thus indirectly activated mTORC1 [142]. Additionally, AKT can also directly activate mTORC1 by phosphorylating and inhibiting PRAS40 (an mTORC1 component mentioned above that negatively regulates complex’s kinase activity) [58,64,75,78,146]. After mTORC1 is activated, a series of cascade effects will be activated to inhibit autophagy. In contrast to mTORC1, Mammucari et al. [147] found that inhibition of mTOR2 in starved mouse skeletal muscle leads to induction of autophagy, which is mediated by the downstream target-FoxO3 transcription factor negatively regulated by AKT. Thus, mTORC2 may also negatively regulate autophagy due to its role in fully activating AKT.

As discussed above, the PI3K/AKT signaling pathway is a well-established upstream regulator of mTOR, which acts to inhibit autophagy by activating mTOR. Previous studies have shown that autophagy not only maintains the balance of liver energy and nutrition, but also plays an important role in regulating liver pathophysiology. Recently, there is increasing evidence that PI3K/AKT/mTOR-mediated autophagy plays a key role in different liver damage and may represents a promising target for novel therapeutic approaches to liver injury.

#### 5.1.1. Liver Fibrosis

As is known to all, liver fibrosis is an important pathophysiological consequence of chronic liver injury [148]. Its main feature is the continuous accumulation of extracellular matrix (ECM), which often leads to the fracture of liver parenchyma through the formation of scar tissue [148]. The harmful effects of fibrosis can be profound and ultimately lead to organ failure. Efforts to prevent fibrotic diseases hinge on the ability to clear excess ECM, overwhelming evidence indicates that hepatic stellate cells (HSCs) are the main source of ECM and its activation is a critical event in the development of liver fibrosis [149]. Therefore, inhibition of HSCs activation is considered to be an important strategy for reversing liver fibrosis [149]. Autophagy plays a double-edged role in liver fibrosis. Most studies have demonstrated that increased autophagy in HSCs can activate HSCs by degrading lipid droplets to provide energy [150,151,152]. For example, Huang et al. found that IGFBPrP1 promotes the activation of HSCs by promoting autophagy mediated by the PI3K/AKT/mTOR signaling pathway [153]. However, Zhang et al. [154] came to different conclusions. They treated HSC-T6 cells with different concentrations of Methyl Helicterate (MH) and found that MH inhibited the activation of HSCs and significantly increased the autophagy level of HSCs. Interestingly, when they used autophagy inhibitors 3-MA or ATG5 knockout to inhibit autophagy in HSC-T6 cells, they found that the ability of MH to inhibit HSC-T6 cells activation was somewhat weaker than before. In contrast, the induction of autophagy with rapamycin increased the apoptosis of HSC-T6 cells. This result suggests that increasing autophagy in HSCs inhibits the activation of HSCs. Further investigation revealed that the PI3K/AKT/mTOR pathway is the main mechanism of MH-induced autophagy in HSCs. MH induces autophagy by inhibiting this pathway. Interestingly, some natural compounds can also play a role in alleviating liver fibrosis through similar pathways as described above. Among them, rutin and curcumin can stimulate the fatty acid-induced autophagy of NHSCs (nonchemical-induced HSC) by regulating the PI3K/AKT/mTOR pathway, and finally found that the activation of NHSCs was inhibited [155]. The above results are all suggest the importance of PI3K/AKT/mTOR-mediated autophagy in reversing liver fibrosis. The reasons for the different results, however, are unclear, probably due to the unique role of the PI3K/AKT/mTOR pathway in the activation of HSCs, or because of the different degrees of autophagy activation in HSCs: moderate activation of HSCs autophagy can provide energy for the activation of HSCs, and excessive activation of HSCs autophagy may cause autophagic cell death of HSCs, thereby inhibiting the activation of HSCs. Further research is still needed. 

#### 5.1.2. Hepatitis B Virus

Hepatitis B virus (HBV) is a small DNA virus with a circovirus that can cause severe liver diseases such as hepatitis, cirrhosis and liver cancer by chronically infecting liver cells. It is reported that about 350 million people worldwide are chronically infected with HBV [156]. Therefore, the mechanism by which hepatitis B virus affects host cells to promote viral replication has been a hot topic of research [156]. Emerging evidence suggests that autophagy is involved in the replication of HBV. For instance, the microRNA-99 family can promote replication of HBV by inducing autophagy [157]. More importantly, the IGF-1R/PI3K/AKT/mTOR/ULK1 signaling pathway was found to be the major mechanism by which the microRNA-99 family regulates autophagy [157]. Additionally, Wang et al. [158] found that hepatitis B virus-x (HBx) transfected Hepg-2 cells (a widely used human hepatoma cell) significantly increased their levels of autophagy through the PI3K/AKT/mTOR pathway. And studies have shown that increasing autophagy in tumor cells can increase their anti-stress ability and promote tumor cell survival [152]. Therefore, PI3K/AKT/mTOR-mediated autophagy may be an important mechanism of liver cancer induced after HBV infection. The above studies indicated that inhibition of PI3K/AKT/mTOR-mediated autophagy may be an emerging target for the treatment of chronic HBV infection. 

#### 5.1.3. Hepatocellular Carcinoma

Hepatocellular Carcinoma (HCC) is a highly malignant and fatal neoplasia. It has been well-documented that autophagy and the PI3K/AKT/mTOR pathway play pivotal regulatory roles in tumorigenesis and cancer therapy, as well as becoming therapeutic targets for many anticancer drugs. Autophagy plays a double-sided role in different stages of HCC development. On the one hand, autophagy can prevent neoplastic transformation by removing damaged organelles and specific proteins in normal cells, and can also limit inflammation and genome instability in cancer cells to exert a cancer suppressing effect [159,160,161]. On the other hand, proper autophagy in established tumor cells can promote the survival and malignant behavior of tumor cells by meeting their metabolic demands, thereby aggravating the deterioration of cancer [162]. Therefore, the significance of balanced autophagy activity in the treatment of cancer has become highly important. At present, autophagy inhibitors chloroquine (CQ) and hydroxychloroquine (HCQ) or autophagy inducer rapamycin and its homologues can play a corresponding role in autophagy regulation at different stages of HCC development [163]. Great progress in the clinical treatment of HCC has been achieved [163]. Moreover, PI3K/AKT/mTOR, an important regulatory pathway upstream of autophagy, also plays a key regulatory role in the development of cancer. Studies have reported that many cancer cells abnormally activate AKT and PI3K, leading to the activation of mTOR, which activates protein synthesis by phosphorylating downstream p70S6K and 4EBP1 to regulate cancer cell proliferation [152]. Therefore, inhibition of the PI3K/AKT/mTOR pathway is a therapeutic target for cancer. For example, some Chinese herbal medicines such as curcumin and sinomenine can exert important anticancer effects by blocking the PI3K/AKT/mTOR pathway [164,165]. However, when the PI3K/AKT/mTOR pathway is inhibited, its key downstream, autophagy, is activated. Activation of autophagy in mature cancer cells promotes malignant behavior, and conflicts with the effects of inhibition of the PI3K/AKT/mTOR pathway on cancer cells. Therefore, the combination of autophagy inhibitors and PI3K/AKT/mTOR pathway inhibitors will make more sense in cancer treatment. For instances, apigenin can inhibit the proliferation of hepatoma cells and induce autophagy by inhibiting the PI3K/AKT/mTOR pathway [166]. Yang et al. [166] found that the combination of autophagy inhibitor and apigenin was more effective. Recently, we have found that PI3K/AKT/mTOR-mediated autophagy plays a different role in HCC than the above. Brusatol activates autophagy of hepatoma cells by inhibiting PI3K/AKT/mTOR pathway, thereby effectively inducing apoptosis of hepatoma cells, inhibiting proliferation of hepatoma cells as well as invasion and migration of tumors [167]. Importantly, activation of autophagy does not promote the malignant behavior of hepatoma cells. The main reason for this result may be that Brusatol over-activates autophagy of liver cancer cells through the PI3K/AKT/mTOR pathway, thereby causing autophagic cell death (also known as programmed cell death (PCD) type II) of liver cancer cells. Moreover, the function of autophagy in carcinogenesis has been controversial. Therefore, our future research should focus more on the role of PI3K/AKT/mTOR-mediated autophagy in different stages of HCC development. Provides more reliable basis for the treatment of HCC by PI3K/AKT/mTOR-mediated autophagy.

### 5.2. AMPK/mTOR/Autophagy Signaling Pathway

Adenosine monophosphate-activated protein kinase (AMPK), a member of the AMPK-associated kinase family, is a highly conserved serine/threonine protein kinase consisting of a catalytic subunit (α subunit) and two regulatory subunits (β subunit and γ subunit) [168] (Figure 4). There are multiple isoforms for each subunit encoded by different genes. In mammals, the catalytic α subunit has two isoforms (α1 and α2), while the β and γ subunits have two (β1 and β2) and three (γ1, γ2 and γ3) isoforms, respectively [169]. AMPK can be composed of any subtype, thus there are 12 potential AMPK combinations under different physiological conditions and these complexes have specific roles [170]. For example, in the muscle, only α1β2γ1, α2β2γ1, and α2β2γ3 complexes are detected. However, after exercise, only the α2β2γ3 complex was activated [171]. This result reveals that the specific subunit composition allows different AMPK complexes to respond to different types of stress stimuli, emphasizing the unique effects of different complexes. Moreover, each subunit has its own unique structure and functions (Figure 4). The C-terminus of the α-subunit contains an autoinhibitory domain (AID) and the N-terminus of the α-subunit contains the kinase domain with a key residue Thr172 [172]. As an important activation site residues of AMPK, some upstream kinases such as the tumor suppressor gene liver kinase B1 (LKB1) and the Ca^2+^/calmodulin-activated protein kinase kinase-2 (CAMKK2, also known as CAMKKβ) can directly activate AMPK by phosphorylating Thr172 [173,174,175,176,177,178]. The β subunit contains a carbohydrate binding module (CBM) that allows AMPK to associate with glycogen [179]. The γ subunit contains four tandem cystathionine-β-synthase (CBS) domains that can bind to AMP or ADP to further act on the α subunit Thr172 site to promote AMPK activation [170,180,181,182]. As the major energy-sensing kinase, AMPK plays an important role in regulating cellular energy homeostasis (Figure 4). AMPK is activated by various types of metabolic stresses or ATP consumption through the mechanisms described above, which involve increases in cellular AMP, ADP or Ca^2+^ and activation of some upstream kinases [183]. Once activated, AMPK maintains energy balance through two complementary actions, inhibiting multiple synthetic pathways in multicellular organisms, such as lipid, protein, and carbohydrate biosynthesis, while activating various catabolic processes such as glucose metabolism and autophagy [169,184].

It is well known that mTOR is a nutrient-sensitive kinase [184]. Under nutrient rich conditions, mTORC1 can integrate various stimuli and signaling networks to promote anabolism, such as stimulating the synthesis of proteins, lipids and nucleotides, while blocking catabolic processes, such as inhibition of autophagy, thereby promoting the growth and proliferation of cells [78,114,185]. Interestingly, this role is exactly the opposite of the effect of AMPK. Therefore, under energy shortage conditions, mTOR is an important downstream target of AMPK. Moreover, AMPK can maintain energy balance by inhibiting the above effects of mTOR and thus playing a vital role in the treatment of some liver injuries. For instance, Quan et al. [186] found that betulinic acid can alleviate non-alcoholic fatty liver by activating the CAMKK/AMPK/mTOR/S6K/SREBP1 pathway. Recently, Hu [187] found that Sestrin 2 can inhibit liver fibrosis by inhibiting rat HSCs activation and proliferation through an AMPK/mTOR-dependent mechanism. The main mechanism of action of the above studies is the inhibitory effect of activated AMPK on some downstream effects of mTOR. 

Notably, autophagy is not only an important downstream regulatory mechanism of mTOR but it also plays a key role in the process of AMPK maintaining energy balance. When cells are stimulated by starvation, autophagy are activated to decompose their own organelles and cytoplasmic components to ensure energy balance. It has been reported that AMPK has an effect of regulating autophagy in both yeast [188] and mammalian cells [189,190]. As mentioned above, ULK1 is an important regulator of the initiation and progression of autophagy and mTOR is the upstream negative regulator of ULK1. Recent studies have shown that AMPK can activate autophagy by indirectly activating ULK1 via the AMPK/mTOR pathway. On the one hand, AMPK inhibits the activity of mTOR by activating the negative regulator of mTORC1, TCS2 [191]. On the other hand, AMPK inhibits the activity of mTOR by inhibiting the major subunit of mTORC1, Raptor [192]. By the action of the above two, the inhibition of ULK1 by mTOR is released, and autophagy is ultimately activated. In addition, AMPK can directly activate ULK1 [110,193] or directly phosphorylate autophagy core components such as ATG9, VPS34 and Beclin-1, to promote autophagy [194,195,196]. There is increasing evidence that AMPK/mTOR-mediated autophagy plays an important role in physiological and pathological conditions. In the next section we will focus on the key role of the AMPK/mTOR/Autophagy signaling pathway in the treatment of various liver diseases.

#### 5.2.1. Non-Alcoholic Fatty Liver Disease

Non-alcoholic fatty liver disease (NAFLD) is the most prevalent chronic liver disease worldwide and is strongly associated with obesity, hyperlipidemia and diabetes [197]. Histologically, the liver of patients with NAFLD will exhibit obvious steatosis, liver inflammation and even hepatocyte necrosis. If uncontrolled, hepatic steatosis will develop into life-threatening diseases such as cirrhosis, hepatocellular carcinoma and liver failure [198,199,200]. Recent investigation revealed that activated autophagy could attenuate liver steatosis. For example, liraglutide (LRG) and Fibronectin type III domain-containing 5 (FNDC5) protein can improve hepatic steatosis and reduces hepatic lipid accumulation by inducing autophagy [201,202]. Conversely, a high-fat diet or long-term accumulation of lipids may reduce autophagy activity [203]. More in-depth research suggests that lipophagy is the main mechanism by which autophagy is involved in regulating lipid metabolism in the liver. The main process of lipophagy involves sequestration of excess lipid drops in autophagosomes, followed by fusion with lysosomes to form autolysosomes, and subsequent degradation by lipidated in the lysosomes [204]. Moreover, it is well documented that increasing AMPK activity is also considered as one of the viable treatment strategies for improving NAFLD [11]. Smith summarized the role of AMPK in influencing NAFLD as the following three main mechanisms. (1) suppression of de novo lipogenesis in liver, (2) increased fatty acid oxidation in the liver, and (3) promotion of mitochondrial function/integrity in adipose tissue [11]. Recently, Guha et al. [205,206] showed that inositol polyphosphate multikinase (IPMK) can interact with AMPK to mediate autophagy via two signal axes, IPMK-AMPK-Sirt-1 and IPMK-AMPK-ULK1. Importantly, they found that deletion of IPMK in cell lines and intact mice almost eliminated lipophagy, promoted liver damage, and impaired hepatocyte regeneration. Thus, IPMK may be a target for NAFLD and liver regeneration therapy. In addition to the above mechanisms, we have recently discovered that activation of AMPK/mTOR regulated autophagy is also an important protective mechanism. For instance, the above mentioned LRG and FNDC5, the main mechanism of its function is to activate AMPK/mTOR-mediated autophagy to improve NAFLD [201,202]. Recently, Shi et al. [207] found that therapeutic dosages of Acetaminophen can aggravate fat accumulation in NAFLD, the potential mechanism might be involved in inhibiting autophagy associated with the AMPK/mTOR pathway. Zhang et al. [208] found that AMPK/mTOR-mediated autophagy levels were significantly inhibited in both high-fat-fed mice and in LO2 cells treated with free fatty acids. However, when Ghrelin o-acyltransferase (GOAT) is inhibited, AMPK/mTOR-mediated autophagy levels are significantly increased and thus liver hepatic toxicity is alleviated. It is suggested that AMPK/mTOR-mediated autophagy activation is an important protective mechanism of NAFLD. Additionally, the therapeutic effect of Hydrogen sulfide (H_2_S) on NAFLD has also been reported. Notably, the main mechanism of action of H_2_S is through activation of AMPK/mTOR-mediated autophagy by acting on AMPKα_2_ [204]. Therefore, activation of AMPK/mTOR-mediated autophagy is an emerging approach to the treatment of NAFLD.

#### 5.2.2. Liver Ischemia and Reperfusion

Ischemia-reperfusion injury is a common complication of patients with partial hepatectomy and liver transplantation, which is divided into two different stages: ischemia and reperfusion. In the ischemic phase, oxygen and nutrient deficiencies can directly lead to liver parenchymal cell damage. During reperfusion, secondary inflammation and oxidative stress aggravate the severity of hepatic ischemia-reperfusion injury [108]. Recent literature suggests that autophagy is a protective mechanism that enables cells to cope with nutrient starvation and hypoxia and improves liver ischemia-reperfusion damage [209,210,211,212]. More importantly, it has recently been discovered that AMPK/mTOR pathway-mediated autophagy plays a key role in the treatment of hepatic ischemia-reperfusion injury in the numerous upstream pathways of autophagy. For example, Kong et al. [212] found that inhibition of GSK3β can improve hepatic ischemia-reperfusion injury by activating autophagy. Subsequent further studies revealed that GSK3β inhibition induces the phosphorylation of AMPK both in liver ischemia-reperfusion and in primary hepatocytes hypoxia/reoxygenation, as well as suppression of mTOR activity. Furthermore, inhibition of AMPK phosphorylation increases mTOR activity and abrogates GSK3β-mediated autophagy against liver ischemia-reperfusion injury. These findings indicate that activation of AMPK/mTOR-mediated autophagy is the main mechanism of inhibition of GSK3β in the treatment of hepatic ischemia-reperfusion injury. Similarly, in liver transplantation models, Liu et al. [213] also demonstrated the pivotal role of AMPK/mTOR-mediated autophagy in hepatic ischemia-reperfusion therapy. Another study [108] showed that isoflurane preconditioning may play a role in the treatment of hepatic ischemia-reperfusion injury by activating hepatocyte autophagy to promote hepatocyte survival and inhibit hepatocyte death/apoptosis. Subsequent further research found that activation of AMPK/mTOR pathway-regulated hepatocyte autophagy is the key to the role of isoflurane. The above findings reflect the important role of AMPK/mTOR-mediated autophagy in the treatment of hepatic ischemia-reperfusion injury. Therefore, AMPK/mTOR-mediated autophagy will be an emerging research target.

#### 5.2.3. Liver Cancer

As mentioned above, in the previous section we have highlighted the double-sided effects of autophagy in liver cancer and demonstrated the important role of PI3K/AKT/mTOR-mediated autophagy in the treatment of liver cancer. Similarly, we have recently discovered that, as another important upstream pathway of autophagy, the AMPK/mTOR pathway also plays an increasingly important role in liver cancer. For instance, as a novel anticancer drug, the main mechanism by which Britannin (Bri) inhibits the growth and proliferation of hepatoma cells is to over-activate AMPK/mTOR pathway-mediated autophagy to promote autophagic cell death [214]. Similarly, BCLB can also treat liver cancer by inducing autophagy to promote autophagic death of cancer cells [215]. Importantly, Liu et al. [215] found that after transfected with Bcl-2-like protein 10 (Bcl2L10/ BCLB), the expression of p-AMPKα (Thr172) and p-Raptor (Ser792) in liver cancer cells was significantly increased, and mTOR (Ser2481) was significantly decreased. These changes underscore that BCLB exerts an anticancer effect by activating AMPK/mTOR-mediated autophagy. Conversely, proper autophagy activation enables cancer cells to survive under stress such as starvation, hypoxia, and drugs [216]. For instance, Glycochenodeoxycholate can promote hepatocellular carcinoma invasion and migration by AMPK/mTOR dependent autophagy activation [217]. It has also been reported that inhibition of autophagy may enhance the sensitivity of cancer cells to cytotoxic drugs [218]. Given that, many autophagy inhibitors have been identified and used in cancer treatment [219]. RA-XII [109], a natural cyclopeptide, can inhibit the growth and colony formation of HepG2 cells. Mechanistic studies suggest that the above effects of RA-XII are mainly by inhibiting autophagy of hepatoma cells. Experiments have confirmed that the AMPK/mTOR/P70S6K pathway plays a central role in RA-XII regulation of autophagy. In summary, the important role of AMPK/mTOR-mediated autophagy in the treatment of liver cancer has been reported. Therefore, another upstream pathway of autophagy, AMPK/mTOR, is also a new target for the treatment of liver cancer.

### 5.3. Ras/Raf/MEK/ERK/mTOR/Autophagy Signaling Pathway

The Ras/Raf/MEK/ERK signaling pathway is an evolutionarily conserved signaling cascade that is normally activated by a variety of extracellular stimuli such as growth factors, hormones, cytokines, and environmental stress [220,221]. Once activated, it triggers a series of phosphorylation events and protein-protein interactions including Ras, Raf, MEK and ERK, which transmit signals from cell surface receptors to promote cell proliferation and survival [220,221,222,223,224] (Figure 3). Ras is a small GTP-binding protein that has been identified in four highly related forms, namely Ha-Ras, N-Ras, Ki-Ras 4A and Ki-Ras 4B [220]. These proteins are usually located in the inner lobules of the plasma membrane [225]. They participate in signal transmission through interaction with multiple effectors and are common upstream molecules of various signal pathways such as Raf/MEK/ERK, PI3K/AKT, RalGDS, PLCε-PKC and Rac1-JNK pathways [225,226]. When cytokines, growth factors or mitogen activate appropriate receptor, the Src homology 2 domain containing protein (Shc) adaptor protein may bind to the C-terminus of the activated specific growth factor receptor [220,227]. Which then binds to the SH2 domain of adaptor protein growth factor receptor binding protein 2 (GRB2) [226]. Notably, GRB2 can recruit Son-of-Sevenless (SOS) into the plasma membrane through its SH3 domain in combination with SOS, forming a Shc/GRB2/SOS complex [226]. Under the stimulation of Shc/GRB2/SOS complex, GDP was separated from Ras protein and replaced by GTP [220,228]. Eventually the Ras undergoes a conformational change, switches from the inactive (GDP-bound) to the active (GTP-bound) form [224].

Raf is a serine/threonine kinase composed of A-Raf, B-Raf and Raf-1 (C-Raf) [220]. Upon Ras activation, Raf is recruited to the cell membrane and promotes Raf dimerization and activation [224]. Downstream of this, activated Raf phosphorylates and activates mitogen-activated protein kinase/ERK kinases 1 and 2 (MEK1/2) [226]. Interestingly, all three members of the Raf family can phosphorylate and activate MEK, but they exhibit different biochemical activities (B-Raf > Raf-1»A-Raf) [220,229]. Following MEK1/2 activation, the only known catalytic substrate, extracellular-signal-regulated kinases 1 and 2 (ERK1/2), was phosphorylated [220,221]. A significant portion of the phosphorylated ERK 1/2 enters the nucleus where it binds and phosphorylates transcriptional regulators to control gene transcription [228,230]. The ERK1/2 signal, however, is not limited to the nucleus, and ERK1/2 can also target substrates outside the nucleus to control metabolism, mitochondrial fission and cell survival [228,231,232]. An increasing number of studies have suggested that Ras/Raf/MEK/ERK signaling pathway plays a role in modulating autophagy. For example, Kim et al. [233] found that abnormal activation of Raf/MEK/ERK increased the expression of important autophagy markers LC3B and SQSTM1/p62 in LNCaP cells, HEK293 cells, BJ cells, and IMR90E1A cells. Growth factor stimulation causes the ERK cascade elements Raf, MEK and ERK to enter the cytoplasmic surface of the autophagosome to interact with the ATG protein [234]. Moreover, Chen et al. [235] also demonstrated that the Ras/Raf/MEK/ERK signaling pathway plays a key role in inhibiting the expression of Cathepsin S to promote autophagy in HONE 1 cells. However, the mechanism by which the Ras/Raf/MEK/ERK pathway is involved in the regulation of autophagy is extremely complex and sometimes seems contradictory. So, at present, our understanding of the role of the Ras/Raf/MEK/ERK pathway in the autophagy environment is limited. In the next section, we will summarize some of the possible mechanisms by which the Ras/Raf/MEK/ERK pathway regulates autophagy.

The key role of Beclin-1 in regulating autophagy has been discussed above. We found that Beclin-1 may be a potential target for the regulation of autophagy by the Ras/Raf/MEK/ERK pathway. For example, the Ras/Raf/MEK/ERK pathway activated in breast cancer MCF-7 cells can induce binding of Noxa (a BH3-only protein) to Mcl-1 (a Bcl-2 family member) such that Beclin1 is derived from Mcl-1 on the dissociation [236]. Similarly, the activated Ras/Raf-1/MEK/ERK pathway in NIH3T3 cells can induce Bcl-2/adenovirus E1B 19-kDa–interacting protein 3 (BNIP3) to induce Bcl-2 release of Beclin1 by increasing the transcriptional expression level of BNIP3 [237]. After Beclin-1 independence, it may bind to the VPS34 protein to form the active center of the core protein phosphatidylinositol triphosphate kinase (class III PI3K) complex during autophagy to promote autophagy. All of the above regulate autophagy through a non-dependent mTOR pathway. Of note, mTOR, the major regulator of autophagy, may also play an important role in the regulation of autophagy in the Ras/Raf/MEK/ERK pathway. It has been reported that activated ERK can activate mTOR1 directly or indirectly. In one aspect, ERK1/2 or its downstream substrate, 90kDa ribosomal S6 kinase (RSK), can positively regulate mTORC1 by inhibiting the activity of the TSC complex [238,239]. On the other hand, ERK1/2 or RSK acts directly on the scaffolding protein Raptor in the mTORC1 complex to activate mTOR [240,241]. In the above, we have discussed in detail how mTOR1 regulates autophagy. So, we hypothesized that the Ras/Raf/MEK/ERK pathway may regulate autophagy by mediating mTOR. Recently, we have surprisingly found that mTOR-dependent autophagy regulated by Ras/Raf/MEK/ERK pathway plays an important role in liver injury.

For example, in the treatment of liver cancer, we have found that many compounds, such as Bicyclol and Azathioprine (AZA), can inhibit the proliferation of hepatoblastoma cells (HepG2) or promote their senescence [242,243]. Wang et al. [242] explored the main mechanism of Bicyclol inhibition of HepG2 cell proliferation, found that the Ras/Raf/MEK/ERK pathway was significantly inhibited, p-mTOR was also inhibited, and LC3-I was converted to LC3-II. These results suggest that Bicyclol acts mainly through activation of Ras/Raf/MEK/ERK/mTOR-mediated autophagy. Similarly, the main mechanism of action of Azathioprine (AZA) in the treatment of liver cancer is through the activation of Ras/Raf/MEK/ERK/TSC2/mTOR-mediated autophagy to promote the senescence of HepG2 [243]. Furthermore, MEK/ERK/mTOR-mediated autophagy also plays a key role in hepatic ischemia-reperfusion injury. However, contrary to the above, Tanshinone IIA and Tri-iodothyronine act to activate autophagy by activating the MEK/ERK pathway to inhibit the activity of mTOR [243,244]. In summary, Ras/Raf/MEK/ERK/mTOR-mediated autophagy does play a key role in liver injury and may be a new research direction for liver injury. However, the specific mechanism of Ras/Raf/MEK/ERK/mTOR regulating autophagy is still unclear, and further research is needed.

### 5.4. Cross-Talk in the Upstream Pathway of the mTOR-Mediated Autophagy

In the previous section, we have separately discussed the detailed mechanisms by which PI3K/AKT, AMPK, Ras/Raf/MEK/ERK pathways regulate autophagy by mediating mTOR. However, they are not three independent parallel pathways. There are multiple crossing points between the three pathways, which affect each other at different stages of signal propagation, both positive and negative, resulting in dynamic and complex cross-talk between the pathways (Figure 5). They synergistically regulate the activity of mTOR to further control autophagy and determine the fate of cells.

#### 5.4.1. Cross-Talk between PI3K/AKT and Ras/Raf/MEK/ERK Pathways

There are significant cross-talks between the kinases of these two pathways under physiological or pathological conditions. PI3K/AKT can affect the Ras/Raf/MEK/ERK pathway at multiple levels [245]. In some circumstances, Ras activation by PI3K has been well characterized. For example, somatostatin receptor [246], insulin and low dose of epidermal growth factor (EGF) [247] can drive Ras activation via PI3K, probably because PI-3,4,5-P_3_ can recruit GAP/Shp2 [248,249]. In addition, the studies have shown that AKT can directly phosphorylate the Ser259 site on Raf-1 and the Ser364 and Ser428 sites on B-Raf in their CR2 amino-terminal regulatory domain [250,251]. Subsequently, the phosphorylated Raf molecule is inactivated by binding to the 14-3-3 adapter protein [250,251,252]. Interestingly, another protein kinase, serum and glucocorticoid-inducible kinase (SGK), which is similar to the AKT catalytic domain, shares a regulatory pathway with AKT [253]. Like AKT, SGK is also directly phosphorylated and activated by PDK1 and can inhibit B-Raf activity by phosphorylating the Ser364 site on B-Raf [253,254,255]. Furthermore, Sato et al. [256] demonstrated that PDK1 not only indirectly regulate Raf by phosphorylation AKT and SGK, but also directly activate MEK by phosphorylation of Ser222 of MEK1 and Ser226 of MEK2 in vivo and in vitro. In human intestinal cells, scholars have found that AKT can indirectly activate ERK [257]. Studies have confirmed that GSK3 plays an important role in this process [257]. Activated GSK3 can inhibit ERK by inhibiting PKCδ [257], which is considered to be an activator of ERK in certain cell lines [258,259]. However, activation of AKT can inhibit activated GSK3 by phosphorylating Ser21 in GSK3α and Ser9 in GSK3β, thus indirectly activating ERK [257,260]. 

In turn, the Ras/Raf/MEK/ERK pathway can also influence the PI3K/AKT pathway through a variety of interaction pathways. Ras-GTP can directly bind and allosterically activate PI3K [261,262,263]. EGF-induced ERK phosphorylates GAB1 on several serine residues near the P85 PI3K binding site, thereby inhibiting GAB1-mediated recruitment of PI3K to EGF receptor (EGFR) [264,265]. Moreover, both ERK and its kinase substrate RSK can regulate PI3K antagonist PTEN by phosphorylating and inhibiting GSK3 [266,267]. Since GSK3 is a negative regulator of PTEN, so the activation of ERK can reduce the PI-3,4,5-P_3_ level and thus inhibit the PI3K/AKT pathway [267]. 

#### 5.4.2. Cross-Talk between PI3K/AKT and AMPK Pathways

AMPK and AKT directly or indirectly regulate mutual phosphorylation to inhibit each other [268]. AMPK is activated upon phosphorylation of Thr172, which is located on the activation loop of the catalytic α-subunit of the kinase and can be phosphorylated by the upstream kinase LKB1 or CaMMK [269]. Phosphorylation of Ser485/491 (equivalent to Ser487/491 in human) inhibits AMPK activity by blocking phosphorylation of Thr172 by LKB1 or CaMMK [270,271,272,273]. Studies have shown that activated AKT can directly phosphorylate Ser485/491 in different cell types, thereby inhibiting AMPK [270,274,275]. Nevertheless, AMPK can also reversely blunt AKT activation. King et al. [276] found that AICAR or phenformin induced the activation of AMPK, which dephosphorylates Ser473 and Thr308 of AKT, thereby inhibiting its activity. In addition, AMPK can also affect the AKT signaling pathway by modulating IRS1. AMPK phosphorylates the Ser794 site of IRS1 (equivalent to the Ser789 site in rats) to inhibit its mediated PI3K/AKT signaling pathway [277,278,279]. However, there are studies indicating that activated AMPK can also stimulate AKT by phosphorylating IRS1 [280,281,282]. It seems that IRS1 has a dual role in the AMPK-mediated AKT signaling pathway, and its function remains to be determined [282,283,284]. Overall, AMPK and AKT have mutual complicated antagonism.

#### 5.4.3. Cross-Talk between AMPK and Ras/Raf/MEK/ERK Pathways

Activated MEK/ERK1/2 pathway can positively regulate AMPK. Zheng et al. [285] found that when the MEK/ERK1/2 pathway was blocked, the phosphorylation of AMPK induced by baicalin was eliminated, while AMPK inhibitors only had little effect on baicalein-induced ERK1/2 phosphorylation. This phenomenon suggests that the MEK/ERK1/2 pathway may activate AMPK. However, ERK can also negatively regulate AMPK. On the one hand, ERK can phosphorylate the Ser485 site of AMPK to prevent phosphorylation of Thr172 site and directly inhibit AMPK activity [286]. On the other hand, ERK can also phosphorylate and inhibit the Ser325 site of LKB1 to indirectly inhibit AMPK [287]. Moreover, the downstream kinase RSK of ERK can also exert a similar effect on AMPK, which exerts an inhibitory effect by phosphorylating the Ser428 site of LKB1 [287].

It is intriguing that AMPK can also affect the Ras/Raf/MEK/ERK pathway. Shen et al. [288] demonstrated that activated AMPK can phosphorylate the Ser729 site of B-Raf. This phosphorylation promotes the binding of B-Raf to the 14-3-3 protein and disrupts the interaction of B-Raf with the KSR1 scaffold protein, ultimately resulting in attenuation of the MEK/ERK signalling [252,288,289]. AMPK can also phosphorylate the Ser621 site on Raf-1 in vitro [290], but subsequent experiments have demonstrated that Raf-1 Ser621 is autophosphorylated in vivo rather than AMPK phosphorylated [291]. In addition, ERK and AMPK are mutually regulated, and many experiments show that AMPK can also reversely activate ERK [292,293,294]. Collectively, growing experiments have shown that AMPK and Ras/Raf/MEK/ERK pathways can be regulated by each other in multiple ways.

#### 5.4.4. Cross-Talk between the Upstream Pathways of mTOR-Mediated Autophagy in Liver Injury

The cross-talk between mTOR-mediated upstream pathways of autophagy is very complex. When we were sorting out the data, we were surprised to find that in some cases, there are multiple pathways together mediate autophagy to regulate liver damage. For example, Bicyclol can inhibit HepG2 cell proliferation by simultaneously inhibiting the PI3K/AKT pathway and the Ras/Raf/MEK/ERK pathway to activate mTOR-mediated autophagy [242]. Moreover, Kaempferol can also activate autophagy of SK-HEP-1 (human hepatic cancer cells) by inhibiting the PI3K/AKT/mTOR pathway or activating the AMPK/mTOR pathway. Excessive autophagy eventually causes autophagic cell death in SK-HEP-1 cell, thereby preventing liver cancer [295]. Although experiments have proved the good therapeutic effect of multiple pathways in liver cancer, the effect of crosstalk between these pathways on liver injury has not been discussed. Therefore, it may be important to clarify the role of crosstalk between mTOR-mediated autophagy upstream pathways on liver injury. These key crosstalk points are likely to be emerging targets for the treatment of liver injury.

## 6. Conclusions and Future Directions

Increasing research indicates that autophagy is a very promising approach to the treatment of liver injury. In the last decade, we have made tremendous strides in understanding the importance of autophagy in the liver. The above summarizes the regulatory effects of autophagy mediated by different drugs through several upstream pathways of mTOR (PI3K/AKT signaling pathway, AMPK signaling pathway and Ras/Raf/MEK/ERK signaling pathway) in various liver injuries (Figure 6), and also found complex crosstalk connection between these three signaling pathways, which are interconnected to collectively regulate mTOR-mediated autophagy. In summary, this review has shown the importance of mTOR upstream pathway in the treatment of liver injury by autophagy, and provides some new research ideas for autophagy in the treatment of liver injury in the future (Table 1).

However, there are still some shortcomings in our understanding of mTOR upstream. Current studies have only proved that autophagy mediated by the upstream pathway of mTOR is involved in the regulation of liver injury, but it is not clear which part of the pathway is regulated by drugs. We only prove that drugs can mediate autophagy through multiple upstream pathways of mTOR in the treatment of liver injury, however, the specific effect of crosstalk between the upstream pathways of mTOR on liver injury is not clear. Therefore, the following research direction should further explore the specific mechanism of drugs targeting the upstream pathway of mTOR, and look for key crosstalk points, which may be a very important target for the treatment of liver injury based on autophagy.

## Figures and Tables

**Figure 1 cells-08-01597-f001:**
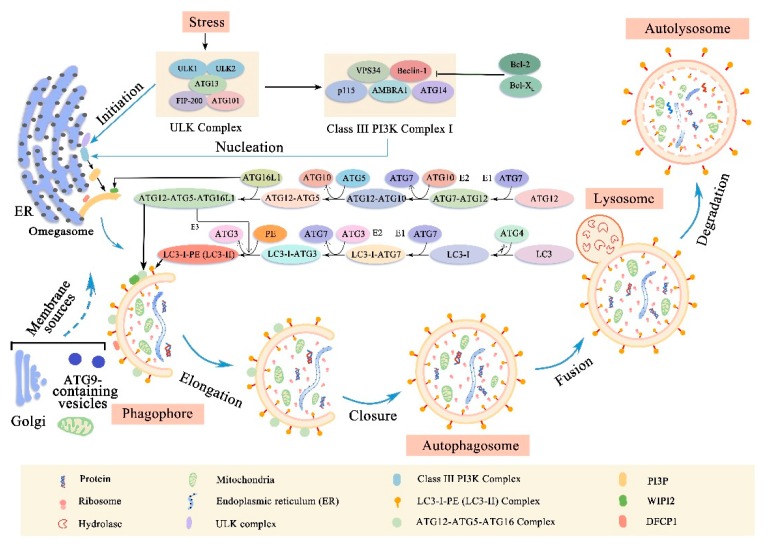
The process and mechanism of autophagy signals that activate the autophagic process (initiation) typically originate from various conditions of stress, such as starvation, hypoxia, oxidative stress, protein aggregation, endoplasmic reticulum (ER) stress and others. The common target of these signaling pathways is the uncoordinated 51-like kinase (ULK) complex (consisting of ULK1/2, ATG13, FIP200 and ATG101) and cause ULK complex translocate to a domain of ER. Upon entry into the ER, the ULK complex initiates nucleation of the phagophore by phosphorylating the class III PI3K complex I (consisting of Beclin-1, PIK3C3/VPS34, PIK3R4/p150, AMBRA1 and ATG14/Barkor). At present, the source of membranes involved in the formation of phagophore remains unclear, possibly ER, mitochondria, Golgi and plasma membrane. After phagophore formation, it will expand and separate damaged organelles (including ER, ribosomes and mitochondria, etc.) and misfolded proteins. As the phagophore expands, the membrane bends, eventually forming a bilayer membrane structure-autophagosome. The ATG12-ATG5-ATg16L1 complex and the LC3-phosphatidylethanolamine (PE) conjugate play an important role during phagophore elongation and closure. Finally, autophagosomes fuse with lysosomes to form autophagosomes and the cargo in autophagosomes is degraded by acid hydrolases in lysosomes. In addition, the nutrients produced by the degradation are released back to the cytoplasm and reused by the cells.

**Figure 2 cells-08-01597-f002:**
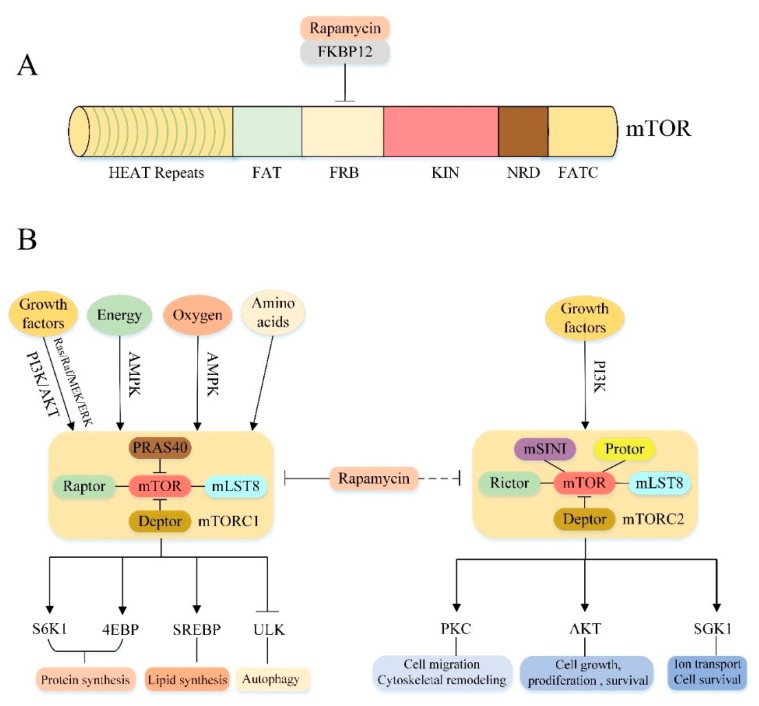
Overview of mTOR signaling pathway (**A**) The primary structure of mTOR. See text for details. (**B**) Molecular composition and upstream and downstream pathways of mTORC1/2: mTOR kinase forms two distinct protein complexes, called mTORC1 and mTORC2, respectively. (Left) mTORC1 consists of three core components (mTOR, Raptor, mLST8) and two inhibitory subunits (PRAS40, Deptor). (Right) mTORC2 consists of four core components (mTOR, Rictor, mLST8, mSIN1) and one inhibitory subunit Deptor. mTORC1 is regulated by growth factors, energy, oxygen and amino acids and is very sensitive to rapamycin. On the one hand, it promotes anabolism by phosphorylation of its substrates S6K1, 4EBP1 and SREBP to accelerate the synthesis of proteins and lipids, on the other hand, it inhibits catabolism by phosphorylation of ULK complex or other ways to prevent autophagy. mTORC2 responds to growth factors; it regulates cell migration, cytoskeletal remodeling, cell growth and proliferation, ion transport, and cell survival through its downstream substrates PKC, AKT, and SGK1. Moreover, mTORC2 is not sensitive to acute rapamycin treatment, but prolonged exposure to the drug can destroys its structure.

**Figure 3 cells-08-01597-f003:**
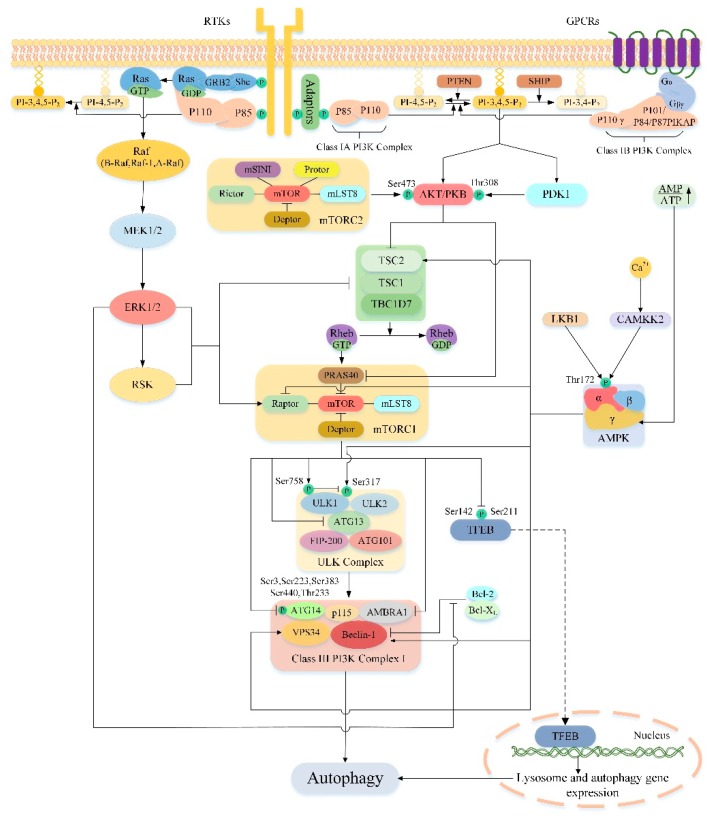
The upstream pathway of mTOR regulates autophagy. The arrows represent the promotion and the blunt arrows represent the inhibition. P represents phosphorylation. See text for details.

**Figure 4 cells-08-01597-f004:**
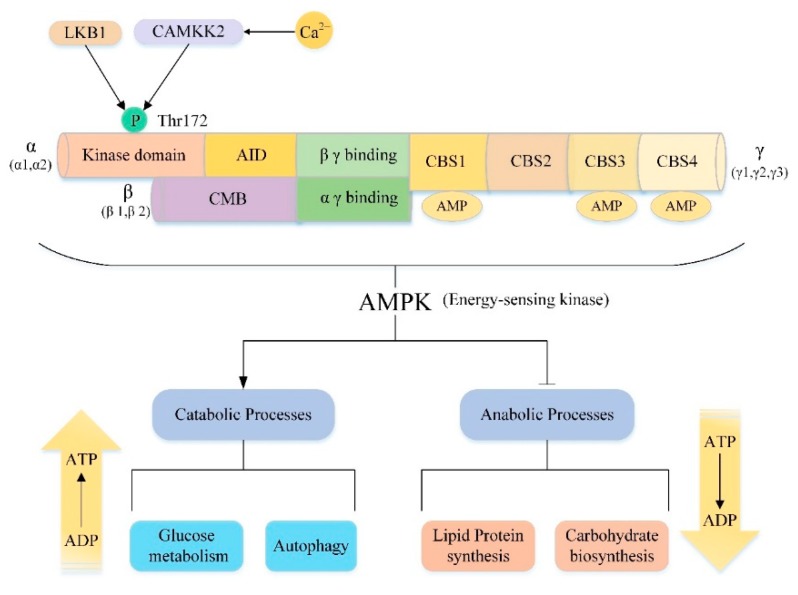
Adenosine monophosphate-activated protein kinase (AMPK) structure and function AMPK are heterotrimeric proteins comprising a catalytic subunit (α subunit) and two regulatory subunits (β subunit and γ subunit), the domain composition of each subunit being shown. Thr172 is a key site for the activation of AMPK. The upstream kinases LKB1 and CAMKK2 are activated by various metabolic stresses and intracellular calcium, respectively. Activation of LKB1 and CAMKK2 activates AMPK by phosphorylation of Thr172. As an energy-sensing kinase, once activated, AMPK inhibits the energy-consuming synthesis process (that is, adenosine triphosphate (ATP) consumption), such as the biosynthesis of lipids, proteins, and carbohydrates, while simultaneously activating the energy-producing process (that is, ATP production), such as glucose metabolism and autophagy to maintain the balance of energy.

**Figure 5 cells-08-01597-f005:**
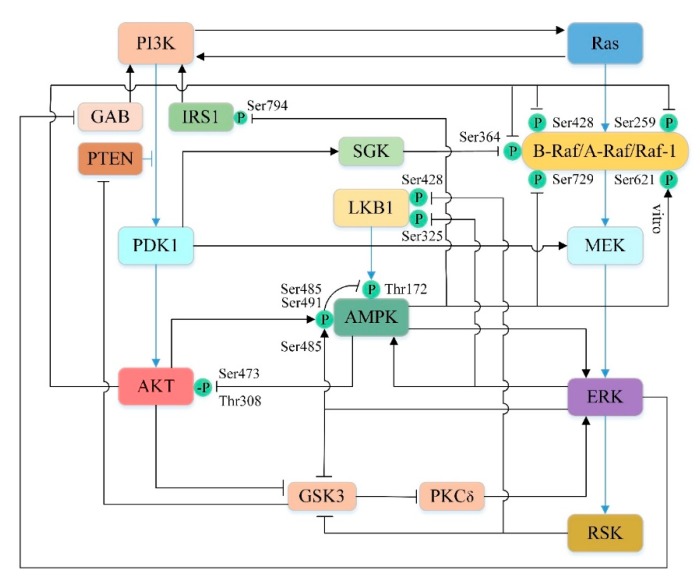
Cross-talk between PI3K/AKT, AMPK and Ras/Raf/MEK/ERK pathway. Black lines indicate crosstalk between the three pathways, with arrows indicating promotion and blunt arrows indicating inhibition. The blue lines indicate the relationship of the three paths themselves. P represents phosphorylation, -P represents dephosphorylation. See text for details.

**Figure 6 cells-08-01597-f006:**
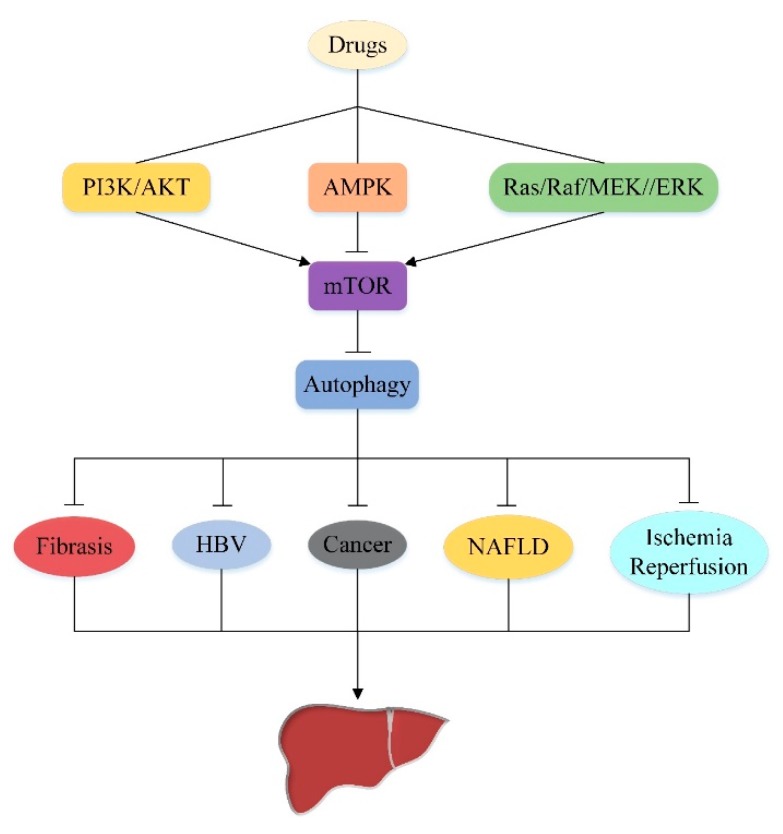
The upstream pathway of mTOR-mediated autophagy in liver injury. PI3K/AKT regulates liver injury by inhibiting mTOR-mediated autophagy. AMPK regulates liver injury by activating mTOR-mediated autophagy. Ras/Raf/MEK/ERK regulates liver injury by inhibiting mTOR-mediated autophagy. PI3K/AKT/mTOR signaling pathway, AMPK/mTOR signaling pathway and Ras/Raf/MEK/ERK/mTOR signaling pathway are emerging targets for drug-based autophagy in the treatment of liver injury.

**Table 1 cells-08-01597-t001:** The role of upstream pathway of mTOR-mediated autophagy in liver disease.

Liver Disease	Signaling Pathway	Effect	Target	Reference
Liver Fibrosis	PI3K/AKT/mTOR/autophagy	IGFBPrP1 promotes the activation of HSCs by activating PI3K/AKT/mTOR-mediated autophagy. MH, Rutin and Curcumin inhibit the activation of HSCs by activating PI3K/AKT/mTOR-mediated autophagy.	Inhibition of PI3K/AKT/mTOR- mediated autophagy Over-activation of PI3K/AKT/ mTOR-mediated autophagy	[153] [154,155]
Hepatitis B virus	PI3K/AKT/mTOR/autophagy	microRNA-99 family can promote replication of HBV by inducing PI3K/AKT/mTOR-mediated autophagy.	Inhibition of PI3K/AKT/mTOR- mediated autophagy	[157]
Non-alcoholic fatty liver disease	AMPK/mTOR/autophagy	LRG, FNDC5 and H_2_S activate AMPK/mTOR-mediated autophagy improve NAFLD. Acetaminophen inhibits AMPK/mTOR-mediated autophagy aggravating fat accumulation in NAFLD.	Activation of AMPK/mTOR- mediated autophagy	[201,202] [204] [207]
Liver ischemia and reperfusion	AMPK/mTOR/autophagy	Isoflurane activates AMPK/mTOR pathway mediated autophagy to improve hepatic ischemia-reperfusion injury. Inhibition of GSK3β can ameliorate hepatic ischemia-reperfusion injury by activating AMPK/mTOR-mediated autophagy.	Activation of AMPK/mTOR- mediated autophagy	[108] [212]
Liver Cancer	PI3K/AKT/mTOR/autophagy AMPK/mTOR/autophagy Ras/Raf/MEK/ERK/ mTOR/autophagy	Brusatol activates PI3K/AKT/mTOR-mediated autophagy of hepatoma cells, thereby effectively inducing apoptosis of hepatoma cells, inhibiting proliferation of hepatoma cells as well as invasion and migration of tumors. Bri and BCLB over-activate AMPK/mTOR-mediated autophagy to inhibit the growth and proliferation of hepatoma cells. Glycochenodeoxycholate activates AMPK/mTOR-mediated autophagy to promote hepatocellular carcinoma invasion and migration. RA-XII inhibits AMPK/mTOR-mediated autophagy to inhibit the growth and colony formation of HepG 2 cells. Bicyclol and AZA activate Ras/Raf/MEK/ERK/ mTOR-mediated autophagy to inhibit the proliferation of HepG2 or promote their senescence.	Over-activation of PI3K/AKT/ mTOR-mediated autophagy Over-activation of AMPK/mTOR- mediated autophagy Inhibition of AMPK/mTOR- mediated autophagy Activation of Ras/Raf/MEK/ERK/ mTOR-mediated autophagy	[167] [214,215] [217] [109] [242,243]

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
