# Peer review of "The Upstream Pathway of mTOR-Mediated Autophagy in Liver Diseases"

_cells, 2019, doi:10.3390/cells8121597_

Round 1
Reviewer 1 Report
The article looks good. However, I encourage to add new findings on autophagy and liver regenrations and lipophagy
Over view the role of lipophagy in liver function Role of autophagy in liver regeneration. Role of IPMK and AMPK in mediating autophagic signaling and liver regenerations.
I believe this points will enrich the review and should be considered.
Author Response
Point 1: The article looks good. However, I encourage to add new findings on autophagy and liver regenrations and lipophagy. Over view the role of lipophagy in liver function Role of autophagy in liver regeneration. Role of IPMK and AMPK in mediating autophagic signaling and liver regenerations. I believe this points will enrich the review and should be considered.
Response 1: Thank you for your comments, your opinion is very helpful to us. We searched two articles on the regulation of autophagy in IPMK on PubMed and supplemented it in the section "5.2.1. Non-alcoholic fatty liver disease". The focus of this review is to describe the role of mTOR-mediated autophagy in liver disease. Therefore, the role of IPMK-mediated autophagy in lipid phagocytosis and liver regeneration has not been discussed extensively.
Reviewer 2 Report
Point 1: Overall, the references in this manuscript were not well cited. Particularly, large previous studies and hallmark review articles were not appropriately cited.
Point 2: The content for the introduction of autophagy must be revised. Many section title and sentences for autophagic process were not correctly described. For example, section 2.1 "Basic morphologic progress" must be changed to " Membrane rearrangement of autophagy process.
Point 3: The authors did not comprehensively update the knowledge of autophagy. The content of section 2.2 and Figure 1 must be revised. For example, not all stresses listed in legend to figure 1 can activte ULK1.....
Point 4: The content of mTOR signaling must be orderly formatted. It did not make sense for the title of section 3.5 "mTOR in liver", must be changed.
Point 5: The content of section 4 "Regulation of autophagy by mTOR" must be strengthened and improved.
Point 6: The title of this manuscript is aim to summarize the role of mTOR-autophagy axis in liver diseases. However, all the aspects of liver diseases were not incorporated into the content of this manuscript. The content of the role of mTOR-regluated autophagy in different kinds of liver diseases must be revised and improved. The summarized table must be provided.
Point 7: The content for the therapeutic target used for treating liver diseases was not well written and summarized.
Author Response
Response to Reviewer 2 Comments
Point 1: Overall, the references in this manuscript were not well cited. Particularly, large previous studies and hallmark review articles were not appropriately cited.
Response 1: Thanks for your comments. We re-examined each cited references and corrected the inappropriate parts. The specific modifications are as follows:
We removed
Guerra, S.; Mamede, A.C.; Carvalho, M.J.; Laranjo, M.; Tralhao, J.G.; Abrantes, A.M.; Maia, C.J.; Botelho, M.F. Liver diseases: what is known so far about the therapy with human amniotic membrane? Cell Tissue Bank 2016 Zhang, J.J.; Meng, X.; Li, Y.; Zhou, Y.; Xu, D.P.; Li, S.; Li, H.B. Effects of Melatonin on Liver Injuries and Diseases. Int J Mol Sci 2017 Herzig, S.; Shaw, R.J. AMPK: guardian of metabolism and mitochondrial homeostasis. Nat Rev Mol Cell Biol 2018 Lavallard, V.J.; Gual, P. Autophagy and non-alcoholic fatty liver disease. Biomed Res Int 2014 Liang, C.; Feng, P.; Ku, B.; Dotan, I.; Canaani, D.; Oh, B.H.; Jung, J.U. Autophagic and tumour suppressor activity of a novel Beclin1-binding protein UVRAG. Nat Cell Biol 2006 Takahashi, Y.; Coppola, D.; Matsushita, N.; Cualing, H.D.; Sun, M.; Sato, Y.; Liang, C.; Jung, J.U.; Cheng, J.Q.; Mule, J.J., et al. Bif-1 interacts with Beclin 1 through UVRAG and regulates autophagy and tumorigenesis. Nat Cell Biol 2007 Ni, H.M.; Williams, J.A.; Yang, H.; Shi, Y.H.; Fan, J.; Ding, W.X. Targeting autophagy for the treatment of liver diseases. Pharmacol Res 2012 Fruman, D.A.; Meyers, R.E.; Cantley, L.C. Phosphoinositide kinases. Annu Rev Biochem 1998 Jiang, B.H.; Liu, L.Z. PI3K/PTEN signaling in angiogenesis and tumorigenesis. Adv Cancer Res 2009 Fruman, D.A.; Chiu, H.; Hopkins, B.D.; Bagrodia, S.; Cantley, L.C.; Abraham, R.T. The PI3K Pathway in Human Disease. Cell 2017We add
Trefts, E.; Gannon, M.; Wasserman, D.H. The liver. Curr Biol 2017 Smith, B.K.; Marcinko, K.; Desjardins, E.M.; Lally, J.S.; Ford, R.J.; Steinberg, G.R. Treatment of nonalcoholic fatty liver disease: role of AMPK. Am J Physiol Endocrinol Metab 2016 Domin, J.; Waterfield, M.D. Using structure to define the function of phosphoinositide 3-kinase family members. FEBS Lett. 1997 Walker, E.H.; Perisic, O.; Ried, C.; Stephens, L.; Williams, R.L. Structural insights into phosphoinositide 3-kinase catalysis and signalling. Nature 1999 Cantley, L.C. The phosphoinositide 3-kinase pathway. Science 2002 Wallace, K.; Burt, A.D.; Wright, M.C. Liver fibrosis. Biochem J 2008
Point 2: The content for the introduction of autophagy must be revised. Many section title and sentences for autophagic process were not correctly described. For example, section 2.1 "Basic morphologic progress" must be changed to "Membrane rearrangement of autophagy process.
Response 2: Thanks for your suggestion. We carefully reviewed some of the references that describe the autophagy process and modified some of the incorrectly titles and sentences in the autophagy process. For example, we have reconfirmed the composition and main functions of class III phosphatidylinositol 3 Kinase (PI3K) complex I, updated the names of each stage of the autophagy process, and changed the title of section 2.2 to "Molecular machinery of autophagy" ... Moreover, we have re-described and supplemented the relevant content and modified the corresponding content in Figure 1 and Figure 3.The stage of nucleation in the autophagy process is the focus of the revision. Could you help us to mark it out if there are other sentences describing the autophagy process are wrong, and we will carefully verify and modify them in time.
Point 3: The authors did not comprehensively update the knowledge of autophagy. The content of section 2.2 and Figure 1 must be revised. For example, not all stresses listed in legend to figure 1 can activte ULK1.....
Response 3: Thanks for your advice. We have read a lot of literature describing the autophagy process in recent years, and updated the content of the initiation, nucleation, fusion and degradation stages of the autophagy process. For example, at initiation we re-described the conditions for activating the ULK complex. During the nucleation phase, we describe in more detail the specific process of phagophore formation, such as "When the class III PI3K complex I is activated, which in turn activates local phosphatidylinositol-3-phosphate (PI3P) production at a characteristic ER structure called the omegasome. From its inception at the omegasome, the phagophore elongates into a cup-shaped structure and begins to engulf cellular material. Then, PI3P recruits the PI3P effector proteins WD repeat domain phosphoinositide-interacting proteins (WIPI2) and zinc-finger FYVE domain-containing protein 1 (DFCP1) to the omegasome.... " During the fusion and degradation stage, we added some proteins involved in fusion and degradation regulation such as "Ras-related proteins in brain small GTPases (such as Rab7 and Rab11), soluble N-ethylmaleimide-sensitive factor attachment protein receptors (SNARES, such as syntaxin-17, SNAP-29, and VAMP8), homotypic fusion and vacuole protein sorting (HOPS) complex components (such as Vps16, Vps33A, and Vps39) ... "Moreover, the content of Figure 1 has been updated accordingly.
Point 4: The content of mTOR signaling must be orderly formatted. It did not make sense for the title of section 3.5 "mTOR in liver", must be changed.
Response 4: Thanks for your comments. The title of section 3.5 has been changed to "The role of mTOR in liver metabolism and liver disease" In the chapter on mTOR signaling pathway, our main purpose is to summarize mTOR, mainly introducing the structural composition of mTOR and the upstream and downstream pathways. The last part of the role of mTOR in the liver metabolism and liver disease is to highlight the theme of the entire article, paving the way for the different upstream of mTOR to play a role in liver disease.
Point 5: The content of section 4 "Regulation of autophagy by mTOR" must be strengthened and improved.
Response 5: Thanks for your comment, your suggestion is very constructive. We reviewed the latest relevant literature and improved Section 4 "mTOR Regulates Autophagy". The emphasis is on complementing the latest downstream subunits of mTOR-regulated autophagy, such as "Cheng et al. Demonstrate that under nutrient-rich conditions, mTORC1 phosphorylates Pacer at serine157 to disrupt the association of Pacer with Stx17 and the HOPS complex and thus abolishes Pacer- mediated autophagosome maturation. Wan et al. reported that mTORC1 can phosphorylates Ser395 of WIPI2, directing WIPI2 to interact specifically with the E3 ubiquitin ligase HUWE1 for ubiquitination and proteasomal degradation. Thereby inhibiting the formation of autophagosome blocks the autophagy flux. "
Point 6: The title of this manuscript is aim to summarize the role of mTOR-autophagy axis in liver diseases. However, all the aspects of liver diseases were not incorporated into the content of this manuscript. The content of the role of mTOR-regluated autophagy in different kinds of liver diseases must be revised and improved. The summarized table must be provided.
Response 6: Thanks for your suggestion. The main purpose of this manuscript is to summarize the role of mTOR's different upstream pathway-mediated autophagy in liver disease. We summarized three different upstream pathways of mTOR and found that not all types of liver disease are related to these three pathways. We just summarize liver diseases that can be mediated by these three pathways and provide a summarized table (Table 1).
Point 7: The content for the therapeutic target used for treating liver diseases was not well written and summarized.
Response 7: Thank you for your comment. We added the Table 1, and summarized in detail the therapeutic targets for liver diseases associated with upstream pathway of mTOR-mediated autophagy. In addition, we discuss the therapeutic targets of liver disease in each section of liver disease. We think this will make the reader understand more thoroughly. Thanks again for your suggestion.

Reviewer 3 Report
The manuscript entitled, "The upstream pathway of mTOR-mediated autophagy in liver diseases" by Zhang and coworkers reviews upstream pathways of mTOR- the PI3K/AKT signaling pathway, the AMPK signaling pathway, and the Ras/Raf/MEK/ERK signaling pathway, in context to their role in liver fibrosis, hepatitis B, non-alcoholic fatty liver, liver cancer, hepatic ischemia reperfusion and other liver diseases through the regulation of mTOR-mediated autophagy. The manuscript is well written and the main ideas are well documented. I would support the publication of this manuscript without further revision.
Author Response
Thank you for your affirmation of our work.
Round 2
Reviewer 1 Report
ACCEPT
Reviewer 2 Report
The authors have strengthened and improved the content of this manuscript. The reviewer suggests the revised manuscript been accepted for publication.